# The metabolic hormone adiponectin affects the correlation between nutritional status and pneumococcal vaccine response in vulnerable indigenous children

Kris E. Siegers[1], Antonius E. van Herwaarden[2], Jacobus H. de Waard[3,4], Berenice del Nogal[5], Peter W. M. Hermans[6], Doorlène van Tienoven[2], Guy A. M. Berbers[7], Marien I. de Jonge[8], Lilly M. Verhagen[1,8,9]*

1 Department of Pediatric Infectious Diseases and Immunology, Wilhelmina Children's Hospital, University Medical Center Utrecht, Utrecht, The Netherlands, 2 Department of Laboratory Medicine, Radboud Institute of Molecular Life Sciences, Radboud University Medical Center, Nijmegen, The Netherlands, 3 Laboratorio de Tuberculosis, Universidad Central de Venezuela, Caracas, Venezuela, 4 One Health Research Group, Facultad de Ciencias de la Salud, Universidad de Las Américas (UDLA), Quito, Ecuador, 5 Department of Pediatrics, Hospital de Niños "J.M. de los Rios", Caracas, Venezuela, 6 Epidemiology Infectious Diseases, Julius Center for Health Sciences and Primary Care, University Medical Center Utrecht, Utrecht, The Netherlands, 7 Center for Infectious Disease Control, National Institute of Public Health and the Environment, Utrecht, The Netherlands, 8 Section of Pediatric Infectious Diseases, Laboratory of Medical Immunology, Radboud University Medical Center, Radboud Center for Infectious Diseases, Nijmegen, The Netherlands, 9 Department of Paediatric Infectious Diseases and Immunology, Radboud University Medical Center, Amalia Children's Hospital, Nijmegen, The Netherlands

* lilly.verhagen@radboudumc.nl

## Abstract

### Background

Almost 200 million children worldwide are either undernourished or overweight. Only a few studies have addressed the effect of variation in nutritional status on vaccine response. We previously demonstrated an association between stunting and an increased post-vaccination 13-valent pneumococcal conjugate vaccine (PCV13) response. In this prospective study, we assessed to what extent metabolic hormones may be a modifier in the association between nutritional status and PCV13 response.

### Methods

Venezuelan children aged 6 weeks to 59 months were vaccinated with a primary series of PCV13. Nutritional status and serum levels of leptin, adiponectin and ghrelin were measured upon vaccination and their combined effect on serum post-vaccination antibody concentrations was assessed by generalized estimating equations multivariable regression analysis.

### Results

A total of 210 children were included, of whom 80 were stunted, 81 had a normal weight and 49 were overweight. Overweight children had lower post-vaccination antibody

**Data Availability Statement:** The relevant data are freely available via the following URL: https://doi.org/10.5061/dryad.0gb5mkm1s.

**Funding:** The original study was supported by Pfizer Venezuela and the Fundacion para la Investigación en Micobacterias, Caracas, Venezuela. None of the authors are affiliated with Pfizer Venezuela. LMV was supported by a Clinical Research Talent fellowship of the UMC Utrecht, The Netherlands. Further, LMV was granted a Pichichero Family Foundation Vaccines for Children Initiative Research Award from the Pediatric Infectious Diseases Society Foundation. The funder did not provide support in the form of salaries for authors and did not have any additional role in the study design, data collection and analysis, decision to publish, or preparation of the manuscript.

**Competing interests:** Pfizer Venezuela only provided financial support in the form of vaccine (PCV13) supply and research materials. Lilly M. Verhagen was supported by a Clinical Research Talent fellowship of the UMC Utrecht, The Netherlands. Further, Lilly Verhagen was granted a Pichichero Family Foundation Vaccines for Children Initiative Research Award from the Pediatric Infectious Diseases Society Foundation. All authors had no significant competing financial, professional or personal interests that might have influenced this research.

**Abbreviations:** BMI, Body Mass Index; GEE, Generalized estimating equations; HAZ, Height-for-age Z-scores; Hib, *Haemophilus influenzae* type b; IFN-γ, Interferon-gamma; IgG, Immunoglobulin G; IL-1, Interleukin-1; IL-6, Interleukin-6; IL-8, Interleukin-8; IL-10, Interleukin-10; LPS, Lipopolysaccharide; NF-kB, Nuclear factor kappa-light-chain-enhancer of activated B cells; PCV7, 7-valent pneumococcal conjugate vaccine; PCV13, 13-valent pneumococcal conjugate vaccine; Th1, T-helper 1; Th2, T-helper 2; TNF, Tumor necrosis factor; WAZ, Weight-for-age Z-scores.

concentrations than normal weight children (regression coefficient -1.15, 95% CI -2.22 —-0.072). Additionally, there was a significant adiponectin-nutritional status interaction. In stunted children, higher adiponectin serum concentrations were associated with lower post-PCV13 antibody concentrations (regression coefficient -0.19, 95% CI -0.24 –-0.14) while the opposite was seen in overweight children (regression coefficient 0.14, 95% CI 0.049–0.22).

## Conclusion

Metabolic hormones, in particular adiponectin, may modify the effect of nutritional status on pneumococcal vaccine response. These findings emphasize the importance of further research to better understand the immunometabolic pathways underlying vaccine response and enable a future of optimal personalized vaccination schedules.

## Introduction

Vaccines are among the most powerful public health achievements in history, preventing debilitating illness and disability, and saving over 2.5 million lives each year [1]. Worldwide, they have lowered both morbidity and mortality associated with infectious diseases. However, there is substantial variation between individuals in the immune response to vaccination. For example, the antibody responses to 7- and 13-valent conjugate pneumococcal (PCV7 and PCV13) and *Haemophilus influenzae* type b (Hib) vaccination vary up to 40-fold in children [2]. This variation has significant consequences for both protective efficacy and duration of protection. Inadequate vaccine responses may leave children at risk of life-threatening infections. An estimated 4 to 19 million children born each year remain unprotected against vaccine-preventable childhood infections, including pertussis, invasive pneumococcal disease and measles, despite receiving routine vaccinations [3]. Several factors, including age, sex, genetics and comorbidities, are known to influence the immune response to vaccination [2]. However, national vaccination schedules and doses are generally the same for every child regardless of these variables. To replace this 'one-size-fits-all' model by more personalized vaccination programmes, insight into the underlying mechanisms and determinants of vaccine responses is needed.

Over the past years, nutritional status has been proposed as an extra factor responsible for the inter-individual variability in protective immunity induced by vaccines. In 2019, 144 million children under the age of five years were affected by chronic undernutrition (stunting, low height-for-age) while 38.3 million children were overweight or obese [4]. In low- and middle-income countries poor nutrition continues to cause nearly half of deaths in children under five, while simultaneously childhood overweight and obesity rates are rising at a rate 30% faster than in higher-income countries [5]. The co-existence of undernutrition alongside overweight and obesity within individuals, households and populations is a double burden of malnutrition [4, 5].

Studies in adults show that an increase in body mass index (BMI) is inversely correlated with antibody responses to Hepatitis A and B vaccines [6–10]. Similarly, a study of trivalent inactivated influenza vaccine immune responses showed a negative correlation between BMI and antibody titres after 12 months [9]. Few studies have been performed in children who are overweight. For hepatitis B and tetanus vaccines, reduced post-vaccination antibody levels

were observed in overweight or obese children [10]. Varying results for vaccination with inactivated influenza vaccines have been reported, with post-vaccination antibody levels in overweight or obese children similar to or slightly higher than in normal weight children of similar age [11, 12]. Undernutrition in relationship to vaccine response has been described mainly in childhood cohorts. Studies assessing post-vaccination antibody levels in children with acute malnutrition (wasting) and/or stunting found mixed results. Severely wasted children are reported to have lower antibody responses to measles and hepatitis B vaccination [13]. For stunting, both negative and positive correlations with antibody levels after PCV vaccination have been described [14, 15].

Both obesity and undernutrition are associated with changes in immune cell number and function that may explain the potential consequences for vaccine response induction. A systematic review on immune alterations in young undernourished children aged <5 years showed that lymphatic tissue, particularly the thymus, undergoes atrophy and cytokine patterns are skewed towards a T-helper 2 (Th2) cell response [13]. Overnutrition resulting in being overweight is thought to cause a chronic state of inflammation with systemic implications for immunity [16]. The adipose tissue is a highly active endocrine organ; responsible for the synthesis and secretion of metabolic hormones [17]. These metabolic hormones and the immune system are known to affect each other [18]. Leptin is a hormone secreted by adipocytes in proportion to adipocyte mass, regulating energy balance and fat stores. Furthermore, leptin also acts as a pro-inflammatory cytokine, linking nutritional status with neuroendocrine and immune functions. The secretion of acute-phase reactants such as interleukin-1 (IL-1) and tumor necrosis factor (TNF) are affected by leptin levels [19]. In addition to leptin, the hormone adiponectin is also secreted by adipocytes. However, in contrast to leptin, the adiponectin production is downregulated in obesity. Adiponectin regulates the expression of several pro- and anti-inflammatory cytokines [20, 21]. Another mediator of fat metabolism is ghrelin. This hormone is produced by the stomach in states of fasting. Immunologically, ghrelin has potent anti-inflammatory activity, inhibiting acute phase protein and pro-inflammatory cytokine expression [22].

Despite the growing body of evidence for the influence of nutritional status on vaccine response, there has been little attention for the physiological explanation of this association. We previously reported that indigenous Venezuelan children suffering from undernutrition (stunting) had higher antibody concentrations following PCV13 vaccination than non-stunted children [14]. Indigenous Venezuelan children are a vulnerable group of children. Although invasive pneumococcal disease has not been studied specifically in this population, a high prevalence rate of acute respiratory tract infections has been described in these children [23]. This has been associated with an increased carriage of *Streptococcus pneumoniae* [24]. Higher pneumococcal carriage rates in indigenous children have also been observed in other populations (e.g. Navajo and White Mountain Apache, Australian Aboriginal and Alaska Native children) and carriage peaks generally co-occur with peaks of disease [25–28]. In unvaccinated indigenous Venezuelan children, vaccine serotypes were predominant [24, 29], underlying the importance of adequate protection by a robust immune response to pneumococcal vaccination.

Here, we present the results of a follow-up study that demonstrates to what extent metabolic hormones may be a modifier in the association between nutritional status and PCV response. To determine the association between pre-vaccination metabolic hormones and post-vaccination antibody levels across the full range of variation in nutritional status, we investigated this relationship in both stunted and overweight children compared with normal weight children.

## Methods

### Study population and setting

The Warao Amerindians live in Antonio Díaz, a municipality located in the Orinoco River Delta in Venezuela that can only be reached by boat. This study included Warao children aged 6 weeks to 59 months from the following nine indigenous communities in Antonio Díaz: Araguabisi, Araguaimujo, Arature, Bonoina, Guayaboroina, Ibaruma, Jobure de Curiapo, Merejina, and Winikina. Door-to-door visits were made to inform all parents of age-eligible children present in these communities during study visits. Children were included between May and November 2012. The original study was registered in a primary registry of the World Health Organization (ICTRP / RPCEC) with identifier number RPCEC00000158. This study included a subset of the original cohort as described in Verhagen et al. 2016 [14], i.e., all children who were stunted or overweight with a serum sample available and an equally large control group of normal weight children. Children who met multiple definitions, i.e. being overweight and stunted, were not included in the subset. Other exclusion criteria were known immunosuppression/deficiency, previous vaccination with any pneumococcal vaccine and major congenital malformations.

### Vaccination schedule

Children aged 6 weeks to 6 months, 7 to 23 months and 24 to 59 months received a primary series of PCV13 including respectively 3, 2 and 1 dose(s) following CDC guidelines [30]. PCV13 contains capsular polysaccharides of 13 serotypes of *Streptococcus pneumoniae* (1, 3, 4, 5, 6A, 6B, 7F, 9V, 14, 18C, 19A, 19F and 23F) conjugated to CRM-197, a non-toxic variant of diphtheria toxin, as carrier protein (Fig 1A).

### Data collection

**Patient characteristics.** Physical examination was performed in all included children, including anthropometric measurements. Pre-vaccination anthropometric measurements

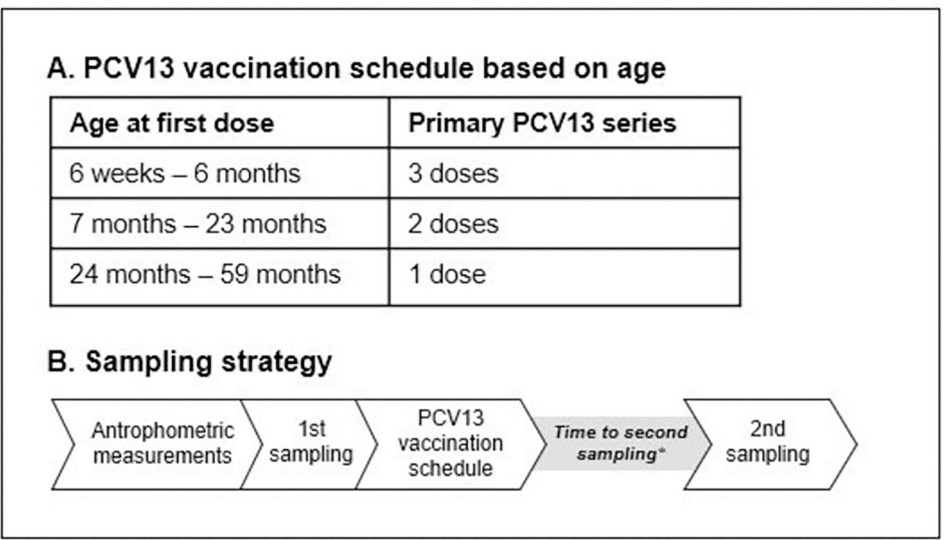

**Fig 1. PCV13 vaccination schedule and sampling strategy. A.** PCV13 vaccination schedule based on age. **B.** Overview of sampling strategy. Blood samples were taken during both the 1st and the 2nd sampling moment. In the pre-vaccination serum samples (1st sampling) PCV13 IgG levels and metabolic hormones were determined. In the post-vaccination serum samples (2nd sampling) only PCV13 IgG levels were determined. * Median 6.7 weeks (IQR 6.4–6.9 weeks).

were transformed into weight-for-age Z-scores (WAZ), height-for-age Z-scores (HAZ) and Body Mass Index (BMI) Z-scores based on WHO standards [31]. Stunting was defined as HAZ <-2 standard deviation (SD). Overweight was defined as a BMI above +1SD. A normal nutritional status was defined as a BMI ≤ +1SD and a HAZ ≥ -2SD.

**Sampling and laboratory methods.** Blood samples were taken just before the first vaccination and again at 1.5 months (median 6.7 weeks (IQR 6.4–6.9 weeks)) after completion of the primary PCV13 vaccination series (Fig 1B). For a detailed description of the sampling and storage procedures we refer to Verhagen et al. 2016 [14]. Determination of pneumococcal serotype-specific serum immunoglobulin G (IgG) concentrations was performed at the National Institute for Public Health and the Environment in Bilthoven, The Netherlands, using a fluorescent bead-based multiplex immunoassay [32].

Metabolic hormones were determined in pre-vaccination serum samples. Serum ghrelin and leptin levels were analysed (in 100 μL) by radioimmunoassay (GHRT-89HK and HL-81K respectively, EMD Millipore Corp.; Missouri USA) as specified by the manufacturer's instructions. Serum adiponectin concentration (High Molecular Weight (HMW) Adiponectin) was analysed (in 10 μL) by chemiluminescence enzyme immunoassay on a Lumipulse analyser G600II (234778, FuijRebio, Gent; Belgium) as specified by the manufacturer's instructions.

## Ethical considerations

Approval by the ethical committee of the Instituto de Biomedicina, Caracas, Venezuela, was granted. In addition, written permission to carry out the study was obtained from the Delta Amacuro Indigenous Health Office and from community leaders of each included community. Children were included upon written informed consent of parents or primary caregivers.

## Statistical analyses

Categorical variables were analysed using Chi-square or Fisher's exact test, as appropriate. For continuous variables, the unpaired Student's t test, nonparametric Mann-Whitney U test or Kruskal Wallis test was used depending on whether the variables were normally distributed (Kolmogorov-Smirnov's test, $p > 0.05$). We used the mean of serotype-specific log-transformed pneumococcal antibody levels as a read-out for pneumococcal vaccine response. Linear regression was performed using the log-transformed antibody concentrations. In our multivariable models we included the following covariates of interest and potential confounders: age (continuous), time from primary series completion until post-vaccination blood sampling (continuous), leptin (continuous), adiponectin (continuous) and ghrelin (continuous, per 20 pg/mL) concentration, community (categorical), mean pre-vaccination antibody response (continuous), BMI (categorical), and HAZ (categorical). Generalized estimating equations (GEE) were used to fit a multivariable linear regression model aimed at identifying possible associations between nutritional status and metabolic hormones and their interactions (independent variables) and post-vaccination log-transformed antibody concentrations (dependant variable), while adjusting for the potential confounders mentioned above. GEEs account for correlation and lack of independence of responses for individuals in clusters within communities using an independence working covariance structure and robust variance estimators. Models were run separately for both stunted vs. non-stunted (i.e. including normal weight and overweight children) and overweight vs. non-overweight children (i.e. including normal weight and stunted children). For all statistical analyses SPSS software version 25 was used. Statistical significance was set to $p$-value $< 0.05$.

**Table 1. Baseline characteristics of the study population.**

| | All (n = 210) | Stunting (n = 80) | Normal weight (n = 81) | Overweight (n = 49) | p-value |
|---|---|---|---|---|---|
| **Sex**, n (%) | | | | | 0.45 |
| Male | 106 (50.5) | 44 (55) | 38 (47) | 22 (45) | |
| Female | 104 (49.5) | 36 (45) | 43 (53) | 27 (55) | |
| **Age** in months (mean, SD) | 35 (16) | 36 (15) | 36 (15) | 31 (17) | 0.27 |
| **Anthropometric measurements**, median | | | | | |
| Weight-for-age Z-scores | -0.57 | -1.81[a] | -0.24[b] | 0.46[c] | <0.01 |
| Height-for-age Z-scores | -1.24 | -2.77[a] | -0.45[b] | -1.26[c] | <0.01 |
| BMI-for-age Z-scores | 0.38 | 0.16[a] | 0.02[a] | 1.54[b] | <0.01 |
| **Metabolic hormones**, median (IQR) | | | | | |
| Leptin (ng/mL) | 7.77 (2.43) | 7.62 (2.25)[a] | 7.54 (2.51)[a] | 8.88 (3.32)[b] | <0.01 |
| Adiponectin (µg/mL) | 6.92 (3.51) | 7.11 (3.67) | 6.37 (3.78) | 6.92 (3.13) | 0.69 |
| Ghrelin (pg/mL) | 684 (307) | 708 (345) | 686 (287) | 630 (328) | 0.17 |

Baseline characteristics are shown for the total study population and groups of stunted, normal weight and overweight children (see methods for the specific definition of the nutritional status groups). Different superscript letters in the same row indicate a significant difference (p <0.05) between the medians. Identical superscript letters in the same row indicate no significant difference between the median values for those categories. IQR = interquartile range.

## Results

The study included 210 children: 80 stunted, 81 normal weight and 49 overweight children. From the 210 samples, insufficient material led to the inability of analysing 16 leptin levels and 7 ghrelin levels. These data points were considered as missing. In samples with sufficient volume, values of leptin, adiponectin and ghrelin were all above measurement detection limits (respectively 0.96 µg/L, 0.09 mg/L and 93 ng/L).

Baseline characteristics of all included children are presented in Table 1. The mean age of included children was 35 months (SD: 16 months). Interestingly, the HAZ score of overweight children was significantly lower than the HAZ score of children with a normal weight (median -1.26 SD vs. -0.45 SD, p <0.01).

Further, significantly higher leptin levels were observed in overweight children compared with stunted and normal weight children (p <0.01). No significant differences in adiponectin or ghrelin levels were observed between children of varying nutritional status.

Only 2.4% of the children had a mean IgG antibody concentration <0.35 µg/ml. For serotype specific IgG concentrations, the number of children with an IgG level <0.35 µg/ml ranged from 0.5–47.1% (S1 Table).

When assessing the relationship between metabolic hormones and vaccine response univariately, an overall positive correlation between both adiponectin and leptin and vaccine response was observed. However, stratification by nutritional status showed that this was only significant in overweight children (Fig 2, Table 2).

In line with our previous results, multivariable analysis showed a trend towards higher antibody levels in stunted compared with non-stunted children (regression coefficient 0.92, 95% CI -0.015–1.85, Table 3). In addition, overweight children appeared to have a significantly lower PCV response compared with non-overweight children (regression coefficient -1.15, 95% CI -2.22 –-0.072). Interestingly, an opposing effect of adiponectin concentrations on post-vaccination antibody levels was observed for stunted vs. overweight children. In stunted children, the interaction of adiponectin levels with stunting was significantly negatively associated with post-vaccination antibody response (regression coefficient -0.19, 95% CI -0.24 –-0.14). In contrast, in overweight children, the interaction of adiponectin levels with

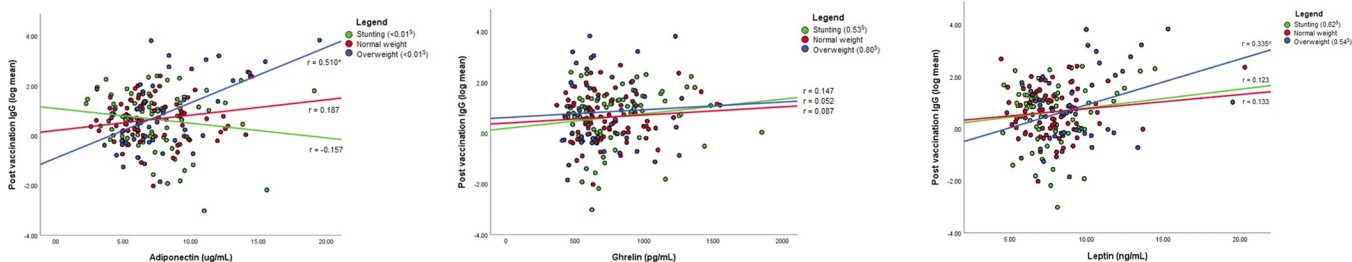

**Fig 2. A: Correlation between adiponectin and post vaccination pneumococcal IgG antibody concentrations.** Correlation, shown by a linear regression line, between pre-vaccination adiponectin levels and post-vaccination log-transformed pneumococcal IgG antibody concentrations for stunted, normal weight and overweight children. r = Pearson correlation coefficient. *p = <0.05. $ P-value value of interaction term in multivariable regression model (Table 3). **B: Correlation between ghrelin and post vaccination pneumococcal IgG antibody concentrations.** Correlation, shown by a linear regression line, between pre-vaccination ghrelin levels and post-vaccination log-transformed pneumococcal IgG antibody concentrations for stunted, normal weight and overweight children. r = Pearson correlation coefficient. $ P-value of interaction term in multivariable regression model (Table 3). **C: Correlation between leptin and post vaccination pneumococcal IgG antibody concentrations.** Correlation, shown by a linear regression line, between pre-vaccination leptin levels and post-vaccination log-transformed pneumococcal IgG antibody concentrations for stunted, normal weight and overweight children. r = Pearson correlation coefficient. *p = <0.05. $ P-value of interaction term in multivariable regression model (Table 3).

overweight was significantly positively associated with post-vaccination antibody levels in multivariable analysis (regression coefficient 0.14, 95% CI 0.049–0.22). Hence, an increase in adiponectin levels in stunted children is associated with lower PCV antibody levels while an increase in adiponectin levels in overweight children is associated with higher PCV antibody levels (Fig 2A, Table 3). Further, we observed a borderline significant association between ghrelin levels and post-vaccination antibody levels in overweight children (regression coefficient 0.009, p = 0.02) (Table 3). No significant association of leptin with PCV13 response was observed in multivariable analysis.

**Table 2. Stratified univariate analysis of the effect of metabolic hormones on vaccine response in children with varying nutritional status.**

| | Serum log mean IgG post-vaccination | | |
|---|---|---|---|
| | **Unadjusted regression coefficient*** | **95% CI** | **p-value** |
| **All children** | | | |
| Leptin (ng/mL) | **0.10** | **0.034–0.17** | **<0.01** |
| Adiponectin (μg/mL) | **0.062** | **0.012–0.11** | **0.016** |
| Ghrelin (per 20 pg/mL) | 0.008 | -0.004–0.020 | 0.19 |
| **Stunted children** | | | |
| Leptin (ng/mL) | 0.071 | -0.059–0.20 | 0.28 |
| Adiponectin (μg/mL) | -0.059 | -0.14–0.023 | 0.16 |
| Ghrelin (per 20 pg/mL) | 0.012 | -0.006–0.029 | 0.20 |
| **Normal weight children** | | | |
| Leptin (ng/mL) | 0.055 | -0.037–0.15 | 0.24 |
| Adiponectin (μg/mL) | 0.062 | -0.009–0.13 | 0.087 |
| Ghrelin (per 20 pg/mL) | 0.006 | -0.010–0.023 | 0.44 |
| **Overweight children** | | | |
| Leptin (ng/mL) | **0.18** | **0.028–0.32** | **0.020** |
| Adiponectin (μg/mL) | **0.22** | **0.12–0.33** | **<0.01** |
| Ghrelin (per 20 pg/mL) | 0.006 | -0.027–0.039 | 0.72 |

A p-value <0.05 is considered statistically significant (in **bold**).

* Linear regression coefficient.

**Table 3. Results from multivariable linear GEE models for post-PCV13 serum antibody levels in stunted and overweight children.**

| | Serum log mean IgG post-vaccination | | |
| --- | --- | --- | --- |
| | Adjusted regression coefficient * | 95% CI | p-value |
| **Undernutrition** | | | |
| Nutritional status (stunted vs. non-stunted) | 0.92 | -0.015–1.85 | 0.054 |
| Metabolic hormones | | | |
| Leptin (ng/mL) | 0.003 | -0.039–0.045 | 0.88 |
| Adiponectin (µg/mL) | **0.076** | **0.018–0.13** | **0.010** |
| Ghrelin (per 20 pg/mL) | 0.007 | -0.013–0.027 | 0.50 |
| Interaction between nutritional status and metabolic hormones | | | |
| Leptin (ng/mL) * stunting | 0.036 | -0.10–0.18 | 0.62 |
| Adiponectin (µg/mL) * stunting | **-0.19** | **-0.24 –-0.14** | **<0.01** |
| Ghrelin (per 20 pg/mL) * stunting | 0.005 | -0.010–0.020 | 0.53 |
| **Overnutrition** | | | |
| Nutritional status (overweight vs. non-overweight) | **-1.15** | **-2.22 –-0.072** | **0.036** |
| Metabolic hormones | | | |
| Leptin (ng/mL) | 0.001 | -0.062–0.064 | 0.97 |
| Adiponectin (µg/mL) | -0.034 | -0.076–0.008 | 0.11 |
| Ghrelin (per 20 pg/mL) | **0.009** | **0.001–0.016** | **0.02** |
| Interaction between nutritional status and metabolic hormones | | | |
| Leptin (ng/mL) * overweight | 0.045 | -0.098–0.19 | 0.54 |
| Adiponectin (µg/mL) * overweight | **0.14** | **0.049–0.22** | **<0.01** |
| Ghrelin (per 20 pg/mL) * overweight | -0.004 | -0.037–0.029 | 0.80 |

Linear regression coefficients were adjusted for the covariates that were included in the regression model in addition to stunting or overweight: age in months, time from primary series completion until post-vaccination blood sampling, leptin, adiponectin, ghrelin, community, mean pre-vaccination antibody response. A p-value <0.05 is considered statistically significant (in **bold**).

* Linear regression coefficient.

## Discussion

The importance of understanding the relationship between nutritional status and vaccine response is underlined by the fact that malnutrition is an increasing global phenomenon. Worldwide, stunting affects around 22% of children under 5 years of age and the rise in childhood obesity prevalence is alarming [33]. In addition, the co-existence of stunting and overweight, i.e. the double burden of malnutrition, is a growing problem since the prevalence of overweight and obesity in many low- and middle income countries has become greater than or equal to prevalence rates observed in high income countries [34]. Vulnerable populations such as indigenous children are even more at risk of developing dual forms of malnutrition [35]. This double burden of malnutrition was also present in our cohort of indigenous Warao children; overweight was accompanied by lower height-for-age Z scores at the individual level. We had previously observed an association between stunting and increased IgG antibody concentrations following PCV13 vaccination [14]. This follow-up study adds the observation that overweight children had a lower PCV13 antibody response. Because of the low number of children with an overall antibody response below the commonly used cut-off of 0.35 µg/ml [36], we cannot comment on the association between nutritional status and a response <0.35 µg/ml. Further, metabolic hormones, particularly adiponectin, appeared to modify the relationship between nutritional status and PCV13 vaccine response.

The relationship between overweight and vaccine response has been studied for hepatitis A, hepatitis B, tetanus and inactivated trivalent influenza vaccination [6, 8–12, 37–40]. Most studies confirm our observation that overweight individuals have lower post-vaccination antibody levels [6, 8–10, 37–40], of which some included children in their study population [10, 37, 40]. In these studies, several explanations were suggested. Chronically elevated levels of interleukin-6 (IL-6) in obese individuals might interfere with the humoral immune system leading to decreased antibody production [37, 41, 42]. Others showed impaired CD8+ T-cell responses in overweight individuals accompanied by less IFN-γ production possibly leading to a decreased vaccination response [9]. Our findings add another explanation for the apparent influence of nutritional status on vaccine response, i.e. the modifying effect of metabolic hormones, particularly adiponectin.

Adiponectin plays an important role in regulating immune responses. By binding to its receptors (AdipoR1 and AdipoR2), expressed on B-cells, it induces secretion of the B-cell derived peptide PEPITEM. This peptide inhibits migration of T cells without affecting recruitment of other leukocytes [20, 43]. Since antibodies are produced by B lymphocytes, the direct effect of adiponectin on those B cells expressing AdipoR may play a role in the mediating effect of adiponectin on the relationship between nutritional status and antibody concentrations. Inflammasomes may also contribute to the mediating effect of adiponectin. Inflammasomes are large multimolecular complexes representing a key signaling platform that triggers inflammatory and immune responses. Adiponectin has shown to inhibit inflammasome activity, thereby suppressing the secretion of pro-inflammatory cytokines. Hence, lower adiponectin levels in obese individuals can lead to increased inflammasome activity and a pro-inflammatory state. However, more research is needed to investigate the exact role of inflammasomes in stunted and overweight individuals and their involvement in vaccine response [44].

In addition, many other signalling pathways might play a role in the effect of adiponectin on vaccine-induced immune response. *In vitro* studies show that adiponectin exposure can promote a transient pro-inflammatory response by the upregulation of pro-inflammatory cytokines, such as TNF-α, IL-6 and IL-8 via activation of the nuclear factor kappa-light-chain-enhancer of activated B cells (NF-B) transcription factor [20, 45, 46]. However, high and prolonged exposure of macrophages to adiponectin leads to an anti-inflammatory milieu by suppression of TNF-α and IL-6 synthesis and induction of IL-10 [20, 46]. In macrophages previously stimulated with lipopolysaccharide (LPS), a less profound inflammatory response is observed upon subsequent exposure, compared with the response upon first exposure, to adiponectin [46, 47]. In stunted children, high levels of LPS have been observed and evidence suggests that chronic LPS exposure may lead to an immunoparalytic state in children with chronic malnutrition [48]. We speculate that increased adiponectin levels in stunted children with an already impaired immune response may contribute to a diminished antibody response upon vaccination.

We observed significantly higher leptin levels in overweight children compared to normal weight and stunted children. This can be explained by the fact that leptin is secreted in proportion to adipocyte mass [19]. Several studies have investigated the immunomodulatory effect of leptin. In both *in vitro* and in murine studies, leptin appears to have a pro-inflammatory effect by inducing the proliferation of mainly naive T cells/T-helper 1 (Th1) cells, while inhibiting the proliferation of memory T cells and Th2 cells [21, 49]. Hence, high levels of leptin in overweight individuals could contribute to a more pro-inflammatory state. However, a study performed in Gambian children aged 7–9 years showed no relationship between leptin and vaccine response [50]. Similarly, our study also suggests that the net effect of leptin levels on pneumococcal conjugate vaccine response is not significant.

Reference values of adiponectin, leptin and ghrelin in healthy individuals are subject to a great variability since there are many influencing factors (among others age, sex and nutritional status). Results of a European study investigating normative values of adiponectin and leptin in children aged <9 years old concluded that the medians of total adiponectin and leptin levels were respectively 8.6–12.7 μg/ml and 1.2–2.2 ng/ml [51]. However, ranges were wide for both hormones and measurement method dependent biases are likely present. Notably, in our study HMW adiponectin was measured with the Lumipulse assay. HMW adiponectin is a fraction of total adiponectin, estimated about 30–50% of total adiponectin [52], nevertheless it has been shown that both total and HMW are interchangeable as they have similar utility when assessing adiponectin levels in blood [52]. Reference values, as supplied by Fuijrebio, of HMW adiponectin in healthy, non-obese adults below 40 years of age in this method are 0.8–9.8 μg/mL for males and 2.1–11.3 μg/mL for females. For ghrelin reference levels in children, only small studies were performed showing an inverse correlation between ghrelin levels and age [53, 54].

Serum sampling in this study took place at a random time during the day; therefore, we cannot exclude an effect of hour-to-hour variation on our study results. However, it seems unlikely that this variation should cause spurious results in a consistent pattern across groups of higher/lower vaccine responses. Since plasma levels of metabolic hormones are affected by the consumption of food during the day [55], consistent differences between stunted and overweight children, related to food consumption patterns, may have occurred. The absence of information on food consumption during the day of sampling is a limitation of this study. However, in general, most indigenous children only eat one meal per day which takes place in the evening, implicating that most children will have been in fasting state during the day.

This study included both stunted and overweight children, all belonging to the indigenous Venezuelan Warao population. Therefore, a possible confounding effect of genetic background or living environment is limited. In addition, we adjusted for possible confounding factors such as age by multivariable analysis to further strengthen the reliability of our results. However, the validity and generalizability of our findings should be substantiated by studying populations in other parts of the world. Moreover, our results are specific for PCV 13 vaccination and may be different for live attenuated or inactivated vaccines. Finally, due to the limited laboratory capacity in these isolated villages, that can only be reached by boat trips of up to >12 hours, we did not have the possibility to further investigate possible causative mechanisms underlying our observations. Therefore, additional research is needed to study the underlying cellular immune mechanisms and to determine whether our findings can be extrapolated to other vaccines.

Because our study shows that nutritional status is an important factor influencing vaccine response, we propose to take these and other factors into account when determining optimal vaccination protocols and strategies. However, current medical practice in vaccinology is to universally administer the same set of vaccines and doses to everyone in the population, in the absence of a contraindication. This approach is based on a national population-level paradigm, allowing the widespread delivery of vaccines, and as a result, the control of many infectious diseases. The major weakness of this approach is that it ignores individual variability in immunologic response and reactogenicity, as well as differences in dose amount needed to generate immunity. The concept of optimizing care by treating patients based on their individual needs and characteristics is increasingly applied in several medical disciplines. Further insights into the underlying mechanisms of the induction of vaccine responses can result in replacement of the 'one-size-fits-all' model by more personalized or sub-population-based vaccination programs that will hopefully yield better immunization results.

In conclusion, this is to our knowledge the first explorative study on the role of metabolic hormones and nutritional status on PCV response in a large cohort of indigenous children. Our results suggest a role for the metabolic hormone adiponectin in vaccine response in children with an altered nutritional status. In stunted children, an increase in adiponectin was associated with a decreased post-vaccination antibody concentration, while the opposite was observed for overweight children. A better understanding of the effect of metabolic hormones on vaccine response in children with an altered nutritional status will support optimal use of existing vaccines and can guide the development of new vaccines with optimal protective efficacy in children with varying nutritional status.

## Supporting information

**S1 Table. Number and percentage of children with a serotype specific post vaccination IgG (μg/ml) response below 0.35 μg/ml, stratified by nutritional status.**
(DOCX)

## Acknowledgments

The authors thank the participating families and the field workers involved in the recruitment and sampling of children, in particular, the medical students of the Escuela de Medicina José Maria Vargas of the Universidad Central de Venezuela and Jochem Burghouts, Meyke Hermsen, Thor Küchler, Stèphan Kraai, and Marcella Overeem. The authors thank Irina Tcherniaeva for technical support.

## Author Contributions

**Conceptualization:** Marien I. de Jonge, Lilly M. Verhagen.

**Data curation:** Kris E. Siegers, Lilly M. Verhagen.

**Formal analysis:** Kris E. Siegers, Lilly M. Verhagen.

**Funding acquisition:** Jacobus H. de Waard, Lilly M. Verhagen.

**Investigation:** Antonius E. van Herwaarden, Doorlène van Tienoven, Guy A. M. Berbers.

**Methodology:** Lilly M. Verhagen.

**Project administration:** Jacobus H. de Waard, Berenice del Nogal.

**Resources:** Antonius E. van Herwaarden, Jacobus H. de Waard, Peter W. M. Hermans, Marien I. de Jonge, Lilly M. Verhagen.

**Supervision:** Jacobus H. de Waard, Berenice del Nogal, Peter W. M. Hermans, Lilly M. Verhagen.

**Visualization:** Kris E. Siegers.

**Writing – original draft:** Kris E. Siegers, Lilly M. Verhagen.

**Writing – review & editing:** Kris E. Siegers, Antonius E. van Herwaarden, Jacobus H. de Waard, Berenice del Nogal, Peter W. M. Hermans, Doorlène van Tienoven, Guy A. M. Berbers, Marien I. de Jonge, Lilly M. Verhagen.

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
