## [Decision Letter · Decision Letter 0]

7 Jul 2021

PONE-D-21-12590

The metabolic hormone adiponectin modifies the association between nutritional status and pneumococcal vaccine response in vulnerable indigenous children

PLOS ONE

Dear Dr. Verhagen,

Thank you for submitting your manuscript to PLOS ONE. After careful consideration, we feel that it has merit but does not fully meet PLOS ONE’s publication criteria as it currently stands. Therefore, we invite you to submit a revised version of the manuscript that addresses the points raised during the review process.

We look forward to receiving your revised manuscript.

Kind regards,

Paolo Magni

Academic Editor

PLOS ONE

Journal Requirements:

2. Please provide the catalog numbers and sources of all kits referred to in lines 177-183.

3. Please provide the registration information for the original clinical study.

4. Please list the names of the nine indigenous Warao communities where participants were from.

5. Thank you for stating the following in the Financial Disclosure section:

[The original study was supported by Pfizer Venezuela and the Fundacion para la Investigación en Micobacterias, Caracas, Venezuela. Lilly M. Verhagen was supported by a Clinical Research Talent fellowship of the UMC Utrecht, The Netherlands.

The funders had no role in study design, data collection and analysis, decision to publish, or preparation of the manuscript.]. 

We note that one or more of the authors have an affiliation to the commercial funders of this research study: Pfizer Venezuela

6. We note that you have indicated that data from this study are available upon request. PLOS only allows data to be available upon request if there are legal or ethical restrictions on sharing data publicly. For information on unacceptable data access restrictions, please see http://journals.plos.org/plosone/s/data-availability#loc-unacceptable-data-access-restrictions.

Reviewers' comments:

Reviewer's Responses to Questions

**Comments to the Author**

1. Is the manuscript technically sound, and do the data support the conclusions?

Reviewer #1: Partly

2. Has the statistical analysis been performed appropriately and rigorously? 

Reviewer #1: No

3. Have the authors made all data underlying the findings in their manuscript fully available?

Reviewer #1: No

4. Is the manuscript presented in an intelligible fashion and written in standard English?

Reviewer #1: Yes

5. Review Comments to the Author

Reviewer #1: 1) Figures 2A-2C only showed the simple regression lines of IgG levels ~ metabolic hormone levels. The data points should also be plotted with different color /shape for the three categories. The significance levels of the interaction terms in the multiple regression models should be reported here to quantitatively show whether the correlations are different between categories.

2) Table 1, it’s still unclear to me how to interpret the different superscript letters. Please be more specific.

3) Table 2, it is unclear the GMRs were between which two groups. For example, for All children category, how was GMR calculated? If it’s a ratio, what are numerator and denominator, respectively? Note that for results in Table 3, it’s clear the comparison is between under-/over-nutrition group and normal group. However, it’s unclear here.

4) Table 3, note that unadjusted GMR are the same for under- and over nutrition groups, and it’s the same as that in Table 2 (1.11, 1.06, and 1.01). Again, please clarify.

5) In the main text, GMR was interpreted as OR, which is incorrect.

6. PLOS authors have the option to publish the peer review history of their article (what does this mean?). If published, this will include your full peer review and any attached files.

Reviewer #1: No

---

## [Author Response · Author response to Decision Letter 0]

20 Sep 2021

Dear dr. Magni,

We thank you for your constructive comments and suggestions. Your comments are highly appreciated, and we have incorporated the provided feedback accordingly. The revised manuscript has been uploaded; please find our response to each specific comment below. 

Journal requirements

1. Please ensure that your manuscript meets PLOS ONE's style requirements, including those for file naming. The PLOS ONE style templates can be found at https://journals.plos.org/plosone/s/file?id=wjVg/PLOSOne_formatting_sample_main_body.pdf and https://journals.plos.org/plosone/s/file?id=ba62/PLOSOne_formatting_sample_title_authors_affilaffilaf.pdf

We have adjusted the manuscript style according to the PLOS ONE’s style requirements. 

2. Please provide the catalog numbers and sources of all kits referred to in lines 177-183.

We have added the catalog numbers and sources of all kits to the ‘sampling and laboratory methods’ section. 

3. Please provide the registration information for the original clinical study. 

The original study was registered in a primary registry of the World Health Organization (ICTRP / RPCEC) with identifier number RPCEC00000158. We have included this in the methods section. 

4. Please list the names of the nine indigenous Warao communities where participants were from.

The nine indigenous Warao communities are named in the revised version of our manuscript. 

5. Thank you for stating the following in the Financial Disclosure section:

[The original study was supported by Pfizer Venezuela and the Fundacion para la Investigación en Micobacterias, Caracas, Venezuela. Lilly M. Verhagen was supported by a Clinical Research Talent fellowship of the UMC Utrecht, The Netherlands. The funders had no role in study design, data collection and analysis, decision to publish, or preparation of the manuscript.]. 

We note that one or more of the authors have an affiliation to the commercial funders of this research study: Pfizer Venezuela

None of the authors has or had an affiliation to Pfizer Venezuela. Jacobus de Waard has received funding from Pfizer Venezuela for the here presented study. However, the funder did not provide support in the form of salaries for authors, nor did they have any additional role in the study design, data collection and analysis, decision to publish, or preparation of the manuscript.

Pfizer Venezuela only provided financial support in the form of vaccine (PCV13) supply and research materials.

The specific roles of the authors are articulated in the ‘author contributions’ section.

There were no competing interests, as clarified above. We have also included this in the updated Funding Statement and Competing Interest Statement in our cover letter. 

6. We note that you have indicated that data from this study are available upon request. PLOS only allows data to be available upon request if there are legal or ethical restrictions on sharing data publicly. For information on unacceptable data access restrictions, please see http://journals.plos.org/plosone/s/data-availability#loc-unacceptable-data-access-restrictions. 

We have uploaded the data in accordance with the provided guidelines and adjusted the text accordingly. The data are now freely available via the following URL: https://datadryad.org/stash/share/LUfQ9WNDrEOw5Aq9YIpEcugbAU7FHfyE0oVmK_abeC4

However, please note that this is a temporary URL because the status of our project is currently set to ‘private for peer review’. We will replace this URL with a definite one upon acceptance of our manuscript.

We have added the captions for our supporting table at the end of our manuscript.

Reviewers’ comments

1. Figures 2A-2C only showed the simple regression lines of IgG levels ~ metabolic hormone levels. The data points should also be plotted with different color /shape for the three categories. The significance levels of the interaction terms in the multiple regression models should be reported here to quantitatively show whether the correlations are different between categories. 

We have adjusted the figures 2A-2C. The figures now also include the plotted data points in addition to the regression lines. The significance levels of the interaction terms in the multiple regression models are added to the legend of the figures. 

2. Table 1, it’s still unclear to me how to interpret the different superscript letters. Please be more specific.

Significantly different median values in the same row are indicated with different superscript letters, while median values that are not significantly different are indicated with identical superscript letters. We have further clarified the interpretation of the superscript letters in the footnote.

3. Table 2, it is unclear the GMRs were between which two groups. For example, for All children category, how was GMR calculated? If it’s a ratio, what are numerator and denominator, respectively? Note that for results in Table 3, it’s clear the comparison is between under-/over-nutrition group and normal group. However, it’s unclear here.

We indeed noted that our presentation of the results was unclear using a GMR. Therefore, we have now used linear regression coefficients for the presentation of the results in tables 2 and 3. 

We have also clarified the comparator groups in the methods section. To avoid over-adjusting, we did not include all malnutrition measures in the same regression model, but instead created two different regression models for undernutrition and overnutrition. We clarified how we analysed malnutrition indicators in the Methods section of the revised manuscript.

4. Table 3, note that unadjusted GMR are the same for under- and over nutrition groups, and it’s the same as that in Table 2 (1.11, 1.06, and 1.01). Again, please clarify.

We thank the reviewer for pointing this out. The unadjusted GMR were the same for all variables, except for nutritional status, because these represented the univariate correlations between e.g. metabolic hormones and antibody concentrations. To be comprehensive, we had initially added these univariate analysis to both tables. However, we acknowledge the unclarity this may have caused and we have now removed the GMR and added linear regression coefficient, in response to the previous comment.

Hence, we have changed tables 2 and 3 so that the stratified univariate analysis is shown in table 2 while table 3 now contains the multivariable linear GEE models.

5. In the main text, GMR was interpreted as OR, which is incorrect.

We thank the reviewer for this observation. Since we now use linear regression coefficients in response to the comments above, we adjusted the text accordingly in the results section. 

We hope that our modifications render our revised manuscript suitable for publication in PLOS ONE.

Sincerely,

On behalf of all authors,

Lilly M. Verhagen

---

## [Decision Letter · Decision Letter 1]

1 May 2022

PONE-D-21-12590R1The metabolic hormone adiponectin modifies the association between nutritional status and pneumococcal vaccine response in vulnerable indigenous childrenPLOS ONE

Dear Dr. Verhagen,

Thank you for submitting your manuscript to PLOS ONE. Before formal acceptance, I invite you to address the minor points raised by Reviewer #2.

We look forward to receiving your revised manuscript.

Kind regards,

Olivier Neyrolles

Section Editor

PLOS ONE

Journal Requirements:

Reviewers' comments:

Reviewer's Responses to Questions

**Comments to the Author**

1. If the authors have adequately addressed your comments raised in a previous round of review and you feel that this manuscript is now acceptable for publication, you may indicate that here to bypass the “Comments to the Author” section, enter your conflict of interest statement in the “Confidential to Editor” section, and submit your "Accept" recommendation.

Reviewer #1: All comments have been addressed

Reviewer #2: All comments have been addressed

2. Is the manuscript technically sound, and do the data support the conclusions?

Reviewer #1: Yes

Reviewer #2: Yes

3. Has the statistical analysis been performed appropriately and rigorously? 

Reviewer #1: Yes

Reviewer #2: Yes

4. Have the authors made all data underlying the findings in their manuscript fully available?

Reviewer #1: Yes

Reviewer #2: Yes

5. Is the manuscript presented in an intelligible fashion and written in standard English?

Reviewer #1: Yes

Reviewer #2: Yes

6. Review Comments to the Author

Reviewer #1: All my comments have been satisfactorily addressed; there is no more comment.

All my comments have been satisfactorily addressed; there is no more comment.

Reviewer #2: This is a very interesting paper addressing a possible role of metabolic hormones and vaccine response, undertaken in a remote indigenous community.

Are there any data on invasive pneumococcal disease in the population studied or other indigenous populations? This would add to the justification for the study & would be useful to know

Minor editorial advice follows below:

Abstract

Line 40: Replace ‘which’ with ‘whom’

Lines 41-2: Replace ‘compared to’ with ‘than’

Introduction

Lines 99-100: Delete ‘described as’

Line 109: Delete ‘observed’

Lines 117-8: Delete ‘in the literature’

Lines 121-2: Replace ‘plays an important role as’ with ‘is’ and delete ‘; it is’

Lines 128-9: Replace ‘the production of adiponectin’ with ‘adiponectin production’

Line 131: Replace ‘is a hormone with’ with ‘has’

Line 137: Replace ‘compared with’ with ‘than’

Methods

Line 147: Delete ‘le’ in reachedle’

Line 191: Why is HMW Adiponectin measured rather than total levels?

Lines 193-7: This is better in the ‘Result’ section

Line 194-7: Replace ‘Values of ….’with ‘In samples with sufficient volume, leptin, adiponectin and ghrelin values were well above…’

Results

Lines 226-7: Delete ‘children’ in ‘80 stunted children, 81 normal weight children and 49 overweight children’.

Discussion

Lines 322-3: Replace ‘Another mechanism that may contribute to the mediating effect of adiponectin is related to the role of inflammasomes’ with ‘Inflammasomes may also contribute to the mediating effect of adiponectin. Inflammasomes are…’

Line 377: Replace ‘the here presented results’ with ‘Our results are specific for PCV 13 vaccination and may be different for live attenuated or inactivated vaccines’

7. PLOS authors have the option to publish the peer review history of their article (what does this mean?). If published, this will include your full peer review and any attached files.

Reviewer #1: No

Reviewer #2: No

---

## [Author Response · Author response to Decision Letter 1]

13 Jun 2022

Dear dr. Neyrolles,

It is a great pleasure to submit to PLOS ONE our revised manuscript entitled ‘The metabolic hormone adiponectin affects the correlation between nutritional status and pneumococcal vaccine response in vulnerable indigenous children’. 

We again thank the Editor and Reviewer for their constructive comments and suggestions. All extra suggestions made by the Reviewer as well as the editorial comments have been implemented in the revised manuscript. The explanatory details are listed below. 

Editor comments

We have meticulously gone through the reference list but could not find any retracted paper. 

We have made one small adjustment to the reference list; in the methods section, we previously referred to the article of Verhagen et al. considering the determination of antibodies using fluorescent bead-based multiplex immunoassay, however, to fully capture the technical details of the assay, we now refer to the article of Stoof et al. 

Following a previous comment of the editor, we have adjusted the title of our manuscript slightly to remove any implication of causality, into 'The metabolic hormone adiponectin affects the correlation between nutritional status and pneumococcal vaccine response in vulnerable indigenous children'.

Reviewers’ comments

1. Are there any data on invasive pneumococcal disease in the population studied or other indigenous populations? This would add to the justification for the study & would be useful to know

Although there are no specific data on invasive pneumococcal disease in our study population, high prevalence rates of acute respiratory tract infections and pneumococcal carriage rates have been observed in this population. These high rates are also seen in other indigenous populations. We have added this as extra information in the introduction section of our revised manuscript. 

2. Minor editorial advices:

We have changed the grammatical adjustments accordingly. Also, we have moved lines 193-7 to the result section. Regarding the question why HMW adiponectin is measured rather than total levels; the Lumipulse assay used for HMW measurements is a very reliable method and it has previously been shown in a comparative analysis that total and HMW adiponectin have similar utility when assessing adiponectin levels in blood (van Andel M, Drent ML, van Herwaarden AE, Ackermans MT, Heijboer AC. A method comparison of total and HMW adiponectin: HMW/total adiponectin ratio varies versus total adiponectin, independent of clinical condition. Clin Chim Acta. 2017;465:30–3).

We are confident that the current manuscript meets the quality criteria of PLOS ONE, and that the subject will be appreciated by the Journal’s readership.

Sincerely,

On behalf of all authors,

Lilly M. Verhagen

---

## [Editor Report · Decision Letter 2]

17 Jun 2022

The metabolic hormone adiponectin affects the correlation between nutritional status and pneumococcal vaccine response in vulnerable indigenous children

PONE-D-21-12590R2

Dear Dr. Verhagen,

We’re pleased to inform you that your manuscript has been judged scientifically suitable for publication and will be formally accepted for publication once it meets all outstanding technical requirements.

Again, we apologize for the delay in processing your manuscript.

Kind regards,

Olivier Neyrolles

Section Editor

PLOS ONE

---

## [Editor Report · Acceptance letter]

12 Jul 2022

PONE-D-21-12590R2 

The metabolic hormone adiponectin affects the correlation between nutritional status and pneumococcal vaccine response in vulnerable indigenous children 

Dear Dr. Verhagen:

I'm pleased to inform you that your manuscript has been deemed suitable for publication in PLOS ONE. Congratulations! Your manuscript is now with our production department. 

Kind regards, 

on behalf of

Dr. Olivier Neyrolles 

Section Editor

PLOS ONE